# Deep Learning with Limited Data: Organ Segmentation Performance by U-Net

**Michelle Bardis** [1,*] , **Roozbeh Houshyar** [1] , **Chanon Chantaduly** [1] , **Alexander Ushinsky** [2] , **Justin Glavis-Bloom** [1] , **Madeleine Shaver** [1] , **Daniel Chow** [1] , **Edward Uchio** [3] and **Peter Chang** [1]

[1]  Department of Radiological Sciences, University of California, Irvine, CA 92617, USA; rhoushya@hs.uci.edu (R.H.); cchantad@hs.uci.edu (C.C.); jglavisb@hs.uci.edu (J.G.-B.); mshaver@hs.uci.edu (M.S.); chowd3@hs.uci.edu (D.C.); changp6@hs.uci.edu (P.C.)

[2]  Mallinckrodt Institute of Radiology, Washington University Saint Louis, St. Louis, MO 63110, USA; aushinsky@wustl.edu

[3]  Department of Urology, University of California, Orange, CA 92868, USA; euchio@hs.uci.edu

\*  Correspondence: mbardis@hs.uci.edu

**Abstract:** (1) Background: The effectiveness of deep learning artificial intelligence depends on data availability, often requiring large volumes of data to effectively train an algorithm. However, few studies have explored the minimum number of images needed for optimal algorithmic performance. (2) Methods: This institutional review board (IRB)-approved retrospective review included patients who received prostate magnetic resonance imaging (MRI) between September 2014 and August 2018 and a magnetic resonance imaging (MRI) fusion transrectal biopsy. T2-weighted images were manually segmented by a board-certified abdominal radiologist. Segmented images were trained on a deep learning network with the following case numbers: 8, 16, 24, 32, 40, 80, 120, 160, 200, 240, 280, and 320. (3) Results: Our deep learning network's performance was assessed with a Dice score, which measures overlap between the radiologist's segmentations and deep learning-generated segmentations and ranges from 0 (no overlap) to 1 (perfect overlap). Our algorithm's Dice score started at 0.424 with 8 cases and improved to 0.858 with 160 cases. After 160 cases, the Dice increased to 0.867 with 320 cases. (4) Conclusions: Our deep learning network for prostate segmentation produced the highest overall Dice score with 320 training cases. Performance improved notably from training sizes of 8 to 120, then plateaued with minimal improvement at training case size above 160. Other studies utilizing comparable network architectures may have similar plateaus, suggesting suitable results may be obtainable with small datasets.

**Keywords:** training size; deep learning; convolutional neural network; U-Net; segmentation; artificial intelligence

## 1. Introduction

Deep learning through convolutional neural networks (CNNs), a subset of artificial intelligence, has demonstrated many strengths for image analysis [1]. For example, CNN approaches represent all recent winning entries within the annual ImageNet Classification challenge, consisting of over one million photographs in 1000 object categories with a 3.6% classification error rate to date [2,3]. In addition, medical applications have demonstrated potential to improve triage with intracranial hemorrhage detection [4] and glioma genetic mutation classification [5]. However, a CNN's performance depends on its ability to learn from the input data itself, and a CNN requires both (1) high-quality and (2) large datasets to solve problems effectively [6,7]. By determining the relationship between

dataset size and CNN accuracy, investigators could potentially calculate when a CNN has been effectively trained.

Training data scarcity and quality are generally not considered challenges for non-biomedical applications where data is widely available. For example, Facebook collects more than 50 TB of video per day and Google processes 200,000 TB per day [8,9]. By contrast, biomedical datasets tend to be heterogenous, difficult to annotate, and relatively scarce [10,11]. In two recent breast imaging studies that used artificial intelligence (AI), the dataset sizes for breast lesion detection and breast cancer recurrence were 320 and 92 patients, respectively [12,13]. Medical studies often lack a combination of publicly available data and high-quality labels [1,14]. Recognition of rare diseases proves especially challenging for medical imaging neural networks, as imaging data for these diseases are often very limited [14]. Additionally, annotation of clinical data is a time consuming and potentially expensive process. Consequently, most medical imaging CNNs face a scarcity of data and calculating an optimal dataset size is infeasible [14].

Since most medical imaging studies are constrained by small datasets, few studies have examined the relationship between the number of cases and CNN performance. A study by Cho et al. [15] compared the number of cases versus performance for a CNN that classified axial computerized tomography scans (CTs) into different anatomic regions: brain, neck, shoulder, chest, abdomen, and pelvis. Another study by Lakhani et al. [16] also observed the performance difference with four different case sizes for CNNs that identified the presence or absence of an endotracheal tube on chest radiographs. Although these two studies showed better accuracy with more cases, the CNNs utilized in the studies completed image classification tasks that make a binary decision after examining the image in its entirety. The relationship between number of cases and segmentation performance within an image has not been rigorously explored.

The purpose of this study is to identify the ideal training size for prostate organ segmentation by analyzing the relationship between the number of MRI cases utilized and consequent CNN performance for imaging analysis. We implemented a type of CNN called a U-Net [17], which was specifically created for medical imaging assessment tasks typically lacking large datasets. U-Net is widely used in medical imaging artificial intelligence (AI) research. We hypothesize a plateau in performance because organ segmentation is a suitable and straightforward task for a U-Net.

## 2. Materials and Methods

### 2.1. Patient Selection

This retrospective study was granted a waiver of informed consent by the institutional review board (IRB) at the University of California, Irvine (UCI) for use of human subject data in a research study. An institutional prostate cancer database was searched to identify patients who had both a (1) prostate multiparametric magnetic resonance imaging (mpMRI) and a (2) magnetic resonance imaging/transrectal ultrasound (MRI/TRUS) fusion biopsy between September 2014 and August 2018. The inclusion criteria for this study included patients who had an mpMRI with subsequent 12-core Artemis 3D TRUS (Eigen, Grass Valley, CA, USA) and MRI/TRUS fusion biopsy using Artemis and ProFuse software (Eigen, Grass Valley, CA, USA) at the University of California, Irvine. An MRI/TRUS fusion biopsy was included as criteria because the prostate organ ground truth was segmented for these patients.

### 2.2. Image Acquisition

The mpMRI images were acquired on a Siemens Magnetom Trio 3-Tesla MRI scanner (Siemens AG, Munich, Germany) and a Phillips Ingenia 3-Tesla MRI scanner (Phillips Healthcare, Amsterdam, Netherlands) at the University of California, Irvine. The image acquisitions were completed in adherence to the prostate imaging reporting and data system (PI-RADS) v2 protocol without endorectal coil (Table 1).

**Table 1.** Magnetic resonance imaging (MRI) acquisition parameters.

| Parameter | Measure |
|---|---|
| Field Strength (B0) | 3 Tesla |
| Acquisition Technique | Turbo spin echo/echo planar |
| Echo Train Length | 25 |
| Time Repetition | 7300 milliseconds |
| Time Echo | 108 milliseconds |
| Flip Angle | 150 degrees |
| Field of View | $200 \times 200$ voxels |
| Matrix Size | $256 \times 205$ pixels |
| Slice Thickness | 3 mm |
| Slice Spacing | 3 mm |
| Coil | Body |

### 2.3. Ground Truth Segmentation

Ten radiologists manually segmented the prostate organ on axial T2-weighted (T2W) images with Profuse software (Eigen, Grass Valley, CA, USA). A board-certified abdominal radiologist with over 10 years of experience (R.H.) was the most experienced radiologist who approved each case. When other radiologists' segmentations differed from his expertise, he refined and updated those segmentations to establish the final ground truth. The mpMRI data and prostate organ segmentation data were transferred to a proprietary research database. From the database, the T2W axial images and organ segmentations were accessed and revised on an in-house image segmenting tool. The in-house tool enabled any segmentation corrections to be completed quickly. This tool integrated with the neural network training software and could be accessed with a web browser. Any segmentation updates were thus seamlessly updated into the neural network implementation.

### 2.4. Image Preprocessing

All axial images were resized to $256 \times 256$ voxels for neural network training. The axial slices were set to have a distance of 3 mm between each other. The standard deviation and mean values of each image were calculated when retrieved from the database. The image signal intensity was then normalized and applied voxelwise to each image. From all the available mpMRI sequences, only the T2W images were used for training and validation.

### 2.5. Convolutional Neural Network

The CNN used for this study was a custom modified U-Net. The algorithm's base architecture was derived from a standard U-Net, which is a fully convolutional contracting and expanding architecture [17]. The customized U-Net has a symmetric architecture and uses the same number of layers during subsampling and upsampling. U-Net also employs skip connections that allow the CNN to combine features for the image contraction and expansion pathways. These skip connections enabled the U-Net to use spatial information that could potentially be lost after the image is further downsampled in the contraction pathway. The entire image was trained during a single forward pass and the U-Net classified each image per pixel.

Our customized U-Net was extended to incorporate three dimensions during training and then produce outputs in two dimensions (Figure 1). Five layers were chosen empirically. In each layer, the image was processed by batch normalization, convolution, rectified linear unit (ReLU) activation, and downsampling with strided convolutions by a factor of 2. The 5 layers used 4, 8, 16, 32, and 32 filters per convolution. The image was downsampled until it became a $1 \times 1 \times 1$ matrix before it underwent expansion. During the expansion pathway, the image was upsampled and a skip connection allowed the upsampled image to combine spatial information from the contraction pathway.

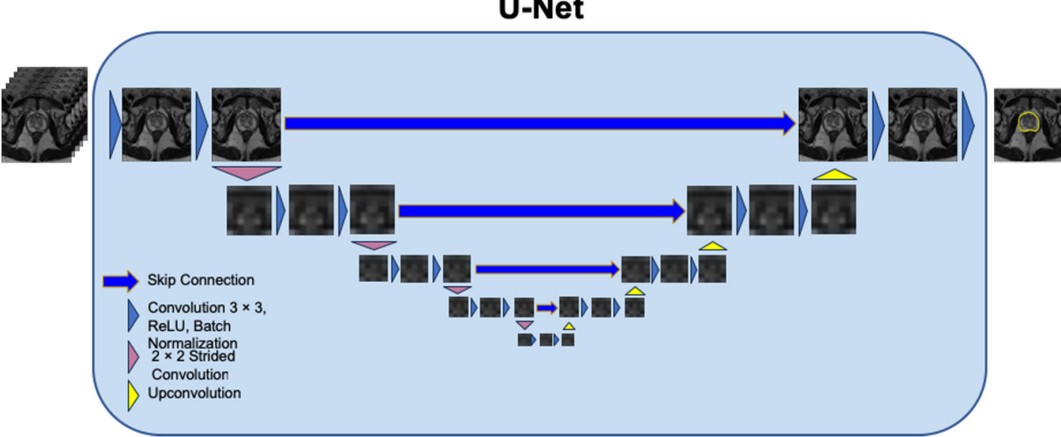

**Figure 1.** All neural network runs were completed on a U-Net with 5 layers. The number of channels used were 4, 8, 16, 32, and 32 for the 5 layers.

### 2.6. Algorithm Training

The Adam optimization algorithm was employed to update the network weights. The Adam algorithm used classical stochastic gradient descent during training [18]. The learning rate was set to $1 \times 10^{-3}$, while the exponential decay rates, $\beta_1$ and $\beta_2$, were set to 0.9 and 0.999, respectively. The batch size was set to 32. The U-Net was trained over a range of iterations: 12,000 to 96,000. The hyperparameters and network structure were kept constant across all 12 runs.

The CNN was written with TensorFlow r1.9 library (Apache 2.0 license) and Python 3.5. The neural network was trained on a graphics processing unit (GPU) workstation which employed four GeForce GTX 1080 Ti cards (11 GB, Pascal microarchitecture; NVIDIA, Santa Clara, CA, USA).

### 2.7. Statistical Methods

The U-Net performance was measured by examining the Dice score. *X* and *Y* are both spatial target regions and their overlap is defined by the Dice score:

$$\text{Dice} = \frac{2 \, | \, X \, \cap \, Y \, |}{|X| + |Y|}. \tag{1}$$

The Dice score quantifies the spatial overlap between the manually segmented and neural network-derived segmentations (Appendix A, Figure A1). A Dice score ranges from 0 (no overlap) to 1 (perfect overlap). A Dice score is the most widely used metric for evaluating segmentation performance for a neural network [19]. To estimate the stability of the neural network during training, the variance of the training Dice score was calculated.

The total number of cases available for training and validation was 400 MRIs. Our U-Net was implemented for 12 runs and trained on the following number of cases: 8, 16, 24, 32, 40, 80, 120, 160, 200, 240, 280, and 320 cases. For each of the 12 runs, the cases were randomly partitioned as either training or validation and the entire set of 400 cases were used. The Dice score was calculated for every validation case in every run. From validation cases in every run, the mean and standard deviation of the Dice scores were computed. For example, the CNN in Run 1 was trained on 8 cases. After the CNN was done training, validation on 392 cases that produced 392 different Dice scores was completed. The mean and standard deviation for these 392 Dice scores were 0.424 and 0.206, respectively. Training size, validation size, mean Dice score, and standard deviation of Dice score are listed for Runs 1 through 12 in Table 2.

**Table 2.** Mean Dice score and standard deviation of Dice score for 12 training sizes.

| Run | Training Size (Cases) | Validation Size (Cases) | Mean Dice Score | Standard Deviation of Dice Score |
|---|---|---|---|---|
| 1 | 8 | 392 | 0.424 | 0.206 |
| 2 | 16 | 384 | 0.653 | 0.160 |
| 3 | 24 | 376 | 0.716 | 0.145 |
| 4 | 32 | 368 | 0.724 | 0.150 |
| 5 | 40 | 360 | 0.747 | 0.147 |
| 6 | 80 | 320 | 0.819 | 0.099 |
| 7 | 120 | 280 | 0.793 | 0.113 |
| 8 | 160 | 240 | 0.858 | 0.068 |
| 9 | 200 | 200 | 0.840 | 0.111 |
| 10 | 240 | 160 | 0.855 | 0.076 |
| 11 | 280 | 120 | 0.857 | 0.082 |
| 12 | 320 | 80 | 0.867 | 0.090 |

After calculating the mean Dice score for 12 different runs, the SciPy [20] library in Python was used to complete curve fitting to these 12 data points with three nonlinear functions (Equations (2)–(4)). Multiple functions were used to optimize the regression and the best three functions that approximate the data were shown (Equations (2)–(4)). These three functions were selected from the SciPy library because they most effectively modeled the dataset that increased quickly from training sizes 8 to 32 and then gradually from training sizes 200 to 320. For all three functions, $a$, $b$, and $c$ were constants, $y$ was the Dice score, and $x$ was the training size (Figure 2). The first function was logarithmic, with the formula:

$$y = a \times \ln(x) + b. \tag{2}$$

The second function was asymptotic and used the formula:

$$y = \frac{a}{b + \frac{1}{x}}. \tag{3}$$

The third function was exponential, with the formula:

$$y = 1 - a \times e^{-b \times x} + c \tag{4}$$

**Mean Dice Score vs. Number of Cases**

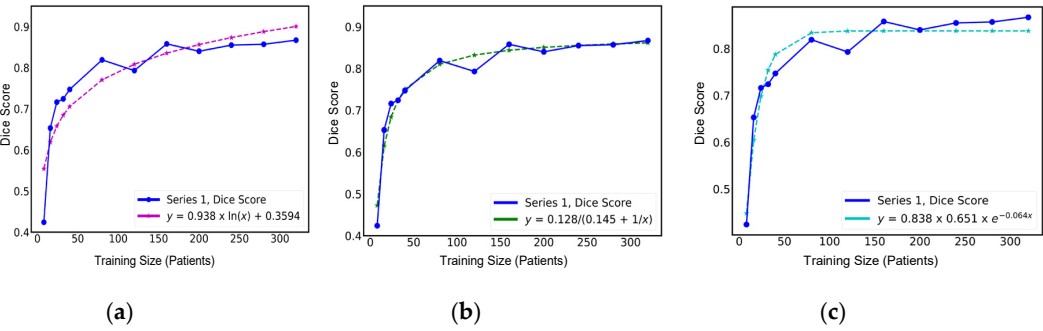

(**a**)  (**b**)  (**c**)

**Figure 2.** The mean Dice score at 12 different training sizes was approximated with several curve functions. (**a**) The first function was logarithmic with the formula $y = a \times \ln(x) + b$. $a$ was 0.938 and $b$ was 0.3594. The mean squared error was $2.55 \times 10^{-3}$. (**b**) The second function was asymptotic and used the formula $y = \frac{a}{b + \frac{1}{x}}$. $a$ was 0.128 and $b$ was 0.145. The mean squared error was $5.70 \times 10^{-4}$. (**c**) The third function was exponential with the formula $y = 1 - a \times e^{-b \times x} + c$. $a$ was 0.651, $b$ was 0.064, and $c$ was −0.162. The mean squared error was $8.30 \times 10^{-4}$. The second function (**b**) provided the best approximation because it had the lowest mean squared error.

For each approximation, the mean squared error was calculated with the following formula:

$$\text{Mean Squared Error} = \frac{1}{n} \sum_{i=1}^{n} (y_i - \widetilde{y_i})^2 \tag{5}$$

where $n$ was 12, $y$ was the Dice score, and $\widetilde{y_i}$ was the estimated Dice score produced by the function.

## 3. Results

### 3.1. Prostate Segmentation

A total of 400 cases (10,400 axial images) from 374 patients were used during training and validation in our study. The average patient age was 65 years (range 41 to 96 years). The average prostate volume was 59 cm$^3$ (range 2 cm$^3$ to 353 cm$^3$). The relationship between number of cases used for training and algorithm performance is shown in Figure 3. The Dice score improved most when the case number changed from 8 to 16 (Table 2). In addition, the Dice score also started to plateau at a training size of 160 cases. The Dice score was 0.858 at 160 cases and 0.867 at 320 cases. To show progression of the Dice score, a single axial image from one case was selected to show the benefits of increasing the number of cases (Figure 4). On this one axial slice, the Dice score progressed from 0 to 0.98 as the training size grew from 8 to 320 cases.

Three nonlinear functions from the SciPy library were used to best fit the mean Dice score performance across the 12 runs. For the first function (2), $a$ was 0.938, $b$ was 0.3594, and the mean squared error was $2.55 \times 10^{-3}$. For the second function (3), $a$ was 0.128, $b$ was 0.145, and the mean squared error was $5.70 \times 10^{-4}$. For the third function (4), $a$ was 0.651, $b$ was 0.064, $c$ was $-0.162$, and the mean squared error was $8.30 \times 10^{-4}$. The best curve fitting was completed by the second function and produced the lowest mean squared error.

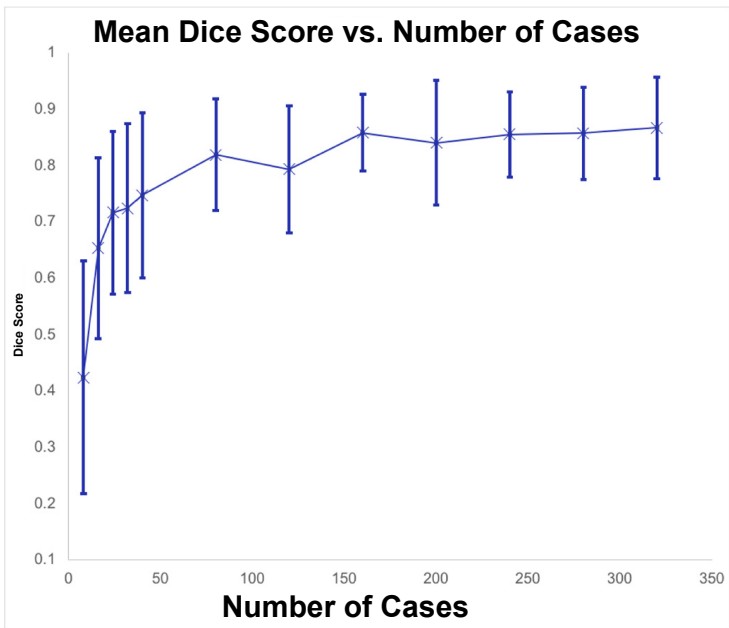

**Figure 3.** Dice score improved the most between 8 cases and 16 cases (0.424 to 0.653). The Dice score started to plateau after 160 cases which had a performance of 0.858. The Dice score only improved by 0.09 from 160 cases to 320 cases. The Dice score was plotted with error bars that show the standard deviation above and below that run's mean Dice score. The standard deviation was lowest at 0.076 with 240 cases and highest at 0.206 with 8 cases.

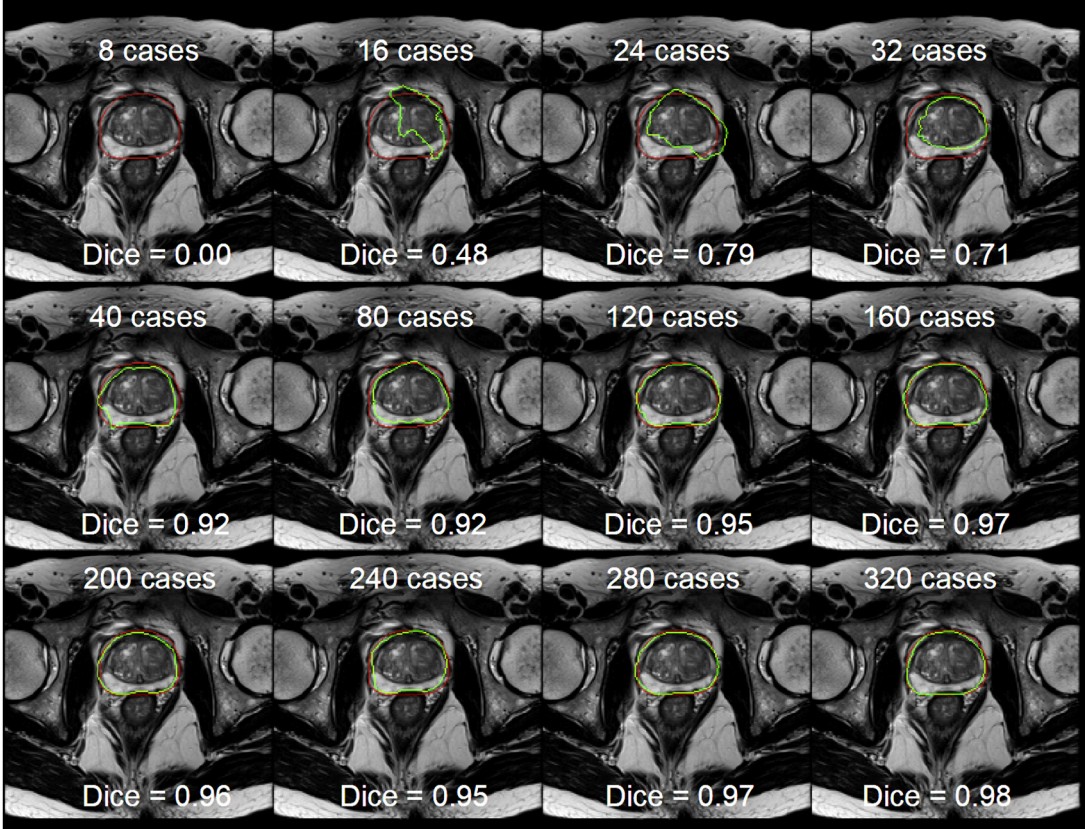

**Figure 4.** The performance of the U-Net was plotted for one axial slice on a single case across the different training sizes. The red line is the ground truth and the green line is the U-Net. The Dice score for one axial slice is shown in each square. The Dice score started to stabilize once the neural network trained with 160 cases.

### 3.2. Convolutional Neural Network Details/Statistics

The stability of the U-Net in training was evaluated (Figure 5). During training, the neural network runs that used training sizes between 8 and 40 did not converge quickly. By contrast, the neural network runs that used training sizes between 80 and 320 did converge quickly. The highest variance was 0.046 for the run that used 40 cases and the lowest variance was 0.003 for that run that used 200 cases. The training process required approximately 7 h of training time for each run. During inference, the U-Net took an average of 0.24 s per case on one GPU to complete inference.

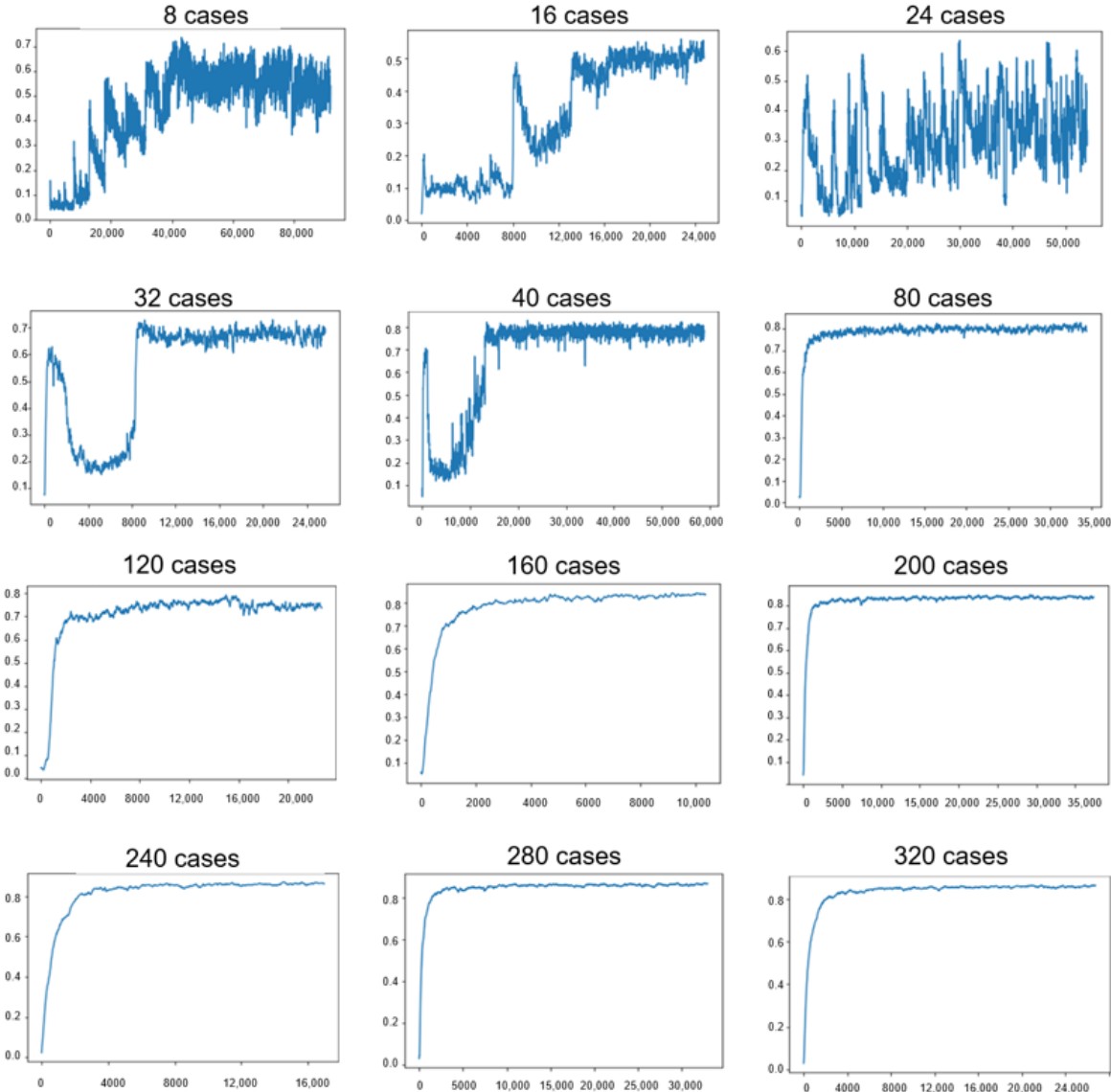

**Figure 5.** The number of iterations is plotted on the *x*-axis and the Dice score during training is plotted on the *y*-axis. The mean Dice score was plotted during training for the 12 different dataset sizes. The Dice score exhibited instability when training on case sizes of 8, 16, 24, 32, and 40. The Dice score stabilized more easily on case sizes of 80, 120, 160, 200, 240, 280, and 320. The Dice score variance was calculated during training; the run with 40 cases had the highest variance of 0.046 and the run with 200 cases had the lowest variance of 0.003.

## 4. Discussion

The purpose of this study was to explore the relationship between training size and CNN performance for prostate organ segmentation. As expected, the CNN performance plateaued with more data after 160 cases, providing a minimal increase in the Dice score. The Dice score was 0.858 at 160 cases and improved to 0.867 at 320 cases. These results confirm our hypothesis that providing more data after a certain size would only provide marginal benefits. The Dice score performance was best modeled with an asymptotic function (Equation (3)) that will converge as the number of cases increases. By using this asymptotic function (Equation (3)) for prediction, the Dice score would reach 0.871 with 500 cases and 0.877 with 1000 cases. The results also demonstrated that the selection of

U-Net as the CNN was apt due to effective prostate segmentation. U-Net's design that classifies each voxel after contraction and expansion are completed to extract unique features make it an apt network for medical imaging analysis. Since manual prostate segmentation is a tedious task [21] and took between 3 and 7 min per case for our radiologists, it is beneficial to know that more cases will not automatically translate into superior results.

Our study is unique because of its dataset size, which enabled us to find an optimal number of cases for training. In ten previous studies that also completed prostate segmentation, the dataset sizes ranged from 21 to 163 cases [22–31]. Three of these studies by Zhu et al. [28], Zhu et al. [27], and Clark et al. [26] were most comparable to our study because they also used a U-Net for their CNN. These three studies obtained Dice scores of 0.89, 0.93, and 0.89 with dataset sizes of 134, 163, and 81 cases, respectively. Although these studies did not compare training with multiple dataset sizes, their results support our findings that U-Net can achieve accurate results for prostate segmentation with a limited dataset.

Along with prostate segmentation, U-Net has demonstrated that it can segment other organs with small dataset sizes. The kidneys were accurately segmented by a U-Net in a study by Jackson et al. [32] with 89 cases. Jackson's study achieved Dice scores of 0.91 and 0.86 for the left and right kidneys, respectively [32]. Multiple U-Nets were combined together to segment multiple organs simultaneously on thorax computed tomography (CT) images in a study by Dong et al. [33]. In Dong's study, the network trained with 40 cases to obtain Dice scores of 0.97, 0.97, 0.90, and 0.87 for the left lung, right lung, spinal cord, and heart, respectively [33]. These studies demonstrate that a U-Net is a well-suited CNN for organ segmentation because of its ability to provide accurate results on small datasets. If these studies were to increase their number of cases, their Dice scores would probably improve and eventually plateau as well.

Several limitations should be considered in our study. All training data were gathered from one academic institution and two manufacturers' MRI scanners. All acquisitions were performed at 3 tesla (3T) MRI field strength and without endorectal coil. Although our CNN works well on our dataset, its ability to generalize with more prostate MRIs outside of our institution could be tested with studies from other institutions. Further work should explore the minimum amount of data for other tasks that build upon prostate organ segmentation. Different dataset sizes could be used to train networks that identify different prostate zones [34] and detect prostate lesions [35]. Along with the prostate, the training dataset size could be varied for other abdominal organs such as the kidney. These studies would serve as useful reference points for future studies that seek to optimize their neural networks. Additional work in this dataset should progress beyond prostate segmentation and detect prostate lesions. Lesion identification is a much more challenging task for AI and data augmentation with a generative adversarial network (GAN) [36] could be very useful since this technical problem lacks sufficient training data [37].

Given the popularity of AI to complete medical imaging projects that perform organ and lesion detection [38], we predict that segmentation projects will likely see diminishing returns in network performance after a threshold number of data points. As such, large datasets may not be a requirement to performing quality AI imaging research. Study teams can start with smaller datasets and evaluate performance analysis on subsets of the training data to predict the plateau effect in their datasets.

## 5. Conclusions

The required number of annotated cases for accurate organ segmentation with a deep learning network may be lower than expected. The marginal benefit of more data may diminish after reaching a threshold number of cases in a deep learning network. In this study of prostate organ segmentation, the U-Net CNN plateaued at 160 cases.

**Author Contributions:** Conceptualization, M.B., D.C., and P.C.; methodology, M.B., and R.H.; software, M.B., C.C., and P.C.; validation, M.B., C.C., and P.C.; formal analysis, M.B., C.C., and P.C.; investigation, M.B., R.H., D.C., and P.C.; resources, D.C., E.U., and P.C.; data curation, M.B., R.H., A.U., J.G.-B., E.U., and P.C.; writing—original

draft preparation, M.B.; writing—review and editing, M.B., R.H., A.U., J.G.-B., M.S., and D.C.; visualization, M.B., R.H., C.C., D.C., and P.C.; supervision, R.H., D.C., and P.C.; project administration, R.H., D.C., and P.C.; funding acquisition, M.B., D.C., and P.C. All authors have read and agreed to the published version of the manuscript.

**Funding:** This research is supported by a Radiological Society of North America Medical Student Research Grant (RMS1902) and additionally by an Alpha Omega Alpha Carolyn L. Kuckein Student Research Fellowship.

**Disclosures:** Author Peter Chang, MD, is a cofounder and shareholder of Avicenna.ai, a medical imaging startup. Author Daniel Chow, MD, is a shareholder of Avicenna.ai, a medical imaging startup.

**Conflicts of Interest:** The authors declare no conflicts of interest. The funders had no role in the design of the study; in the collection, analyses, or interpretation of data; in the writing of the manuscript, or in the decision to publish the results.

## Appendix A

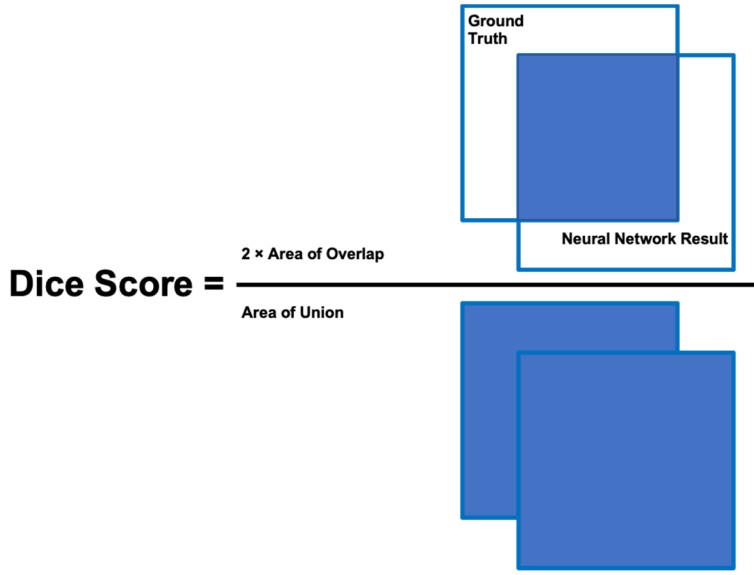

**Figure A1.** The Dice score is used to measure the performance of the neural network. Its range is from 0 (worst) to 1 (best). A score of 1 demonstrates perfect overlap between the ground truth and the neural network's output. A score of 0 shows that the ground truth and neural network's output have no intersection.

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
