# Peer review of "Deep Learning with Limited Data: Organ Segmentation Performance by U-Net"

_electronics, doi:10.3390/electronics9081199_

Round 1

Reviewer 1 Report

The purpose of this study is to identify the ideal training size for prostate organ segmentation by analyzing the relationship between the number of cases and CNN performance for imaging analysis.

The subject of the study is very relevant and highly topical. The paper is well structured but needs to be improved for acceptance. The remarks are as follows:

1. Acronyms should be introduced before they are used. (IRB, MRI, …)

2. Put the reference (15) just after the first appearance of the word U-Net (ligne 65 : We implemented a type of CNN called a U-Net [15], which was specifically ….…)

3. Why the choice of th U-Net?

4. Section 2.3. Ground Truth Segmentation : were there any divergences among the ten radiologists?

5. Please introduce the overall index before Table 2.

6. Is the value of overall index table 2 on learning or test group? is it the mean value? what is the standard deviation?

7. Section 2.7. Statistical Methods is not relevant and should be rewritten. Why do we have to approximate Dice score? Why the choice of functions 2, 3, 4? Introduce the parameters X, Y of equation 1. Equation 5, MSE must be divided by the number of samples n.

8. Each value of the blue curve in Figure 2 is the mean value ? Would it be more relevant to approximate the mean value with standard deviations (as shown in Figure 3)!

9. Is an error in the title of figure 2: the learning rate of the CNN … ?

10. wouldn't it make more sense to redefine the dice score by the following formula:

 (X intersect Y) / X

11. how the variance of the training Dice score is calculated?

12. Figure 5, is it the mean value ? what about the standard deviations?

13. please find these references to argue your article:

***** Application of the deep learning in the biomedical field :

A. Deep Learning in the Biomedical Applications: Recent and Future Status, April 2019, Applied Sciences 9(8):1526, DOI: 3390/app9081526

**** Some examples of the use of small databases in the bio-medical field:

A. Prediction of Oncotype DX recurrence score using deep multi-layer perceptrons in estrogen receptor-positive, HER2-negative breast cancer, May 2020, Breast Cancer, DOI: 1007/s12282-020-01100-4

B. Constructive Deep Neural Network for Breast Cancer Diagnosis, January 2018, DOI: 1016/j.ifacol.2018.11.660

Reviewer 2 Report

The authors present an investigation into the effects of training data size on  the ability of a U-Net to "learn" how to segment MRI images of prostate organs. They demonstrate that smaller amounts of data are required than may be expected for performance to plateau.

The paper is well structured and presented, with clear logical and well supported conclusions.

This reviewer has the following recommendations:

Ln19 - What is IRB?

Ln56 "A study by Cho et al. [13] compared the number of cases versus performance for a CNN" : briefly outline their findings!

Ln119: This paragraph describes the data partitioning. Was the data partitioning randomised on each of the 12 runs?

Ln145: " The Dice score growth rate was approximated with three different functions " Can you justify why more than one is needed, and why you have settled on 3? Figure 2 helps show this. but it would be good to expand on this a little here

Reviewer 3 Report

This study focused on not general deep learning but U-net, therefore the title has to include "U-net".

The authors did not use the data augmentation technque. Please demonstrate the effect of data augmentation such as translation and rotation in the small dataset.

How about the training time for each experiment?

Can this experiment be extended to other generative deep learning technqiues such as GAN? Please discuss about this issue.

Round 2

Reviewer 3 Report

The manuscript is revised and very informative.